# Healthcare Utilization and Costs in Sepsis Survivors in Germany–Secondary Analysis of a Prospective Cohort Study

**DOI:** 10.3390/jcm11041142

**Published:** 2022-02-21

**Authors:** Konrad F. R. Schmidt, Katharina Huelle, Thomas Reinhold, Hallie C. Prescott, Rebekka Gehringer, Michael Hartmann, Thomas Lehmann, Friederike Mueller, Konrad Reinhart, Nico Schneider, Maya J. Schroevers, Robert P. Kosilek, Horst C. Vollmar, Christoph Heintze, Jochen S. Gensichen

**Affiliations:** 1Institute of General Practice and Family Medicine, Jena University Hospital, D-07743 Jena, Germany; katharinahuelle@gmx.de (K.H.); rebekka.gehringer@med.uni-jena.de (R.G.); 2Center of Sepsis Control and Care (CSCC), Jena University Hospital, D-07747 Jena, Germany; 3Institute of General Practice and Family Medicine, Charité University Medicine, D-10117 Berlin, Germany; christoph.heintze@charite.de; 4Institute of Social Medicine, Epidemiology and Health Economics, Charité University Medicine, D-10117 Berlin, Germany; thomas.reinhold@charite.de; 5Department of Internal Medicine, University of Michigan, Ann Arbor, MI 48109-5368, USA; hprescot@med.umich.edu; 6VA Center for Clinical Management Research, Ann Arbor, MI 48105, USA; 7Hospital Pharmacy, Jena University Hospital, D-07747 Jena, Germany; michael.hartmann@med.uni-jena.de; 8Institute of Medical Statistics, Information Sciences and Documentation, Jena University Hospital, D-07747 Jena, Germany; thomas.lehmann@med.uni-jena.de; 9Thiem-Research GmbH, Carl-Thiem-Klinikum, D-03048 Cottbus, Germany; fr.mueller@asklepios.com; 10Department of Anaesthesiology and Intensive Care Medicine, Charité University Medicine Berlin, D-10117 Berlin, Germany; konrad.reinhart@charite.de; 11Institute of Psychosocial Medicine and Psychotherapy, Jena University Hospital, D-07743 Jena, Germany; nico.schneider@med.uni-jena.de; 12Department of Health Sciences, University Medical Center Groningen, University of Groningen, NL-9700 AB Groningen, The Netherlands; m.j.schroevers@umcg.nl; 13Institute of General Practice and Family Medicine, University Hospital of the Ludwig-Maximilians-University Munich, D-80336 Munich, Germany; robert_philipp.kosilek@med.uni-muenchen.de (R.P.K.); jochen.gensichen@med.uni-muenchen.de (J.S.G.); 14Department of Family Medicine, Ruhr-University Bochum Medical School, D-44801 Bochum, Germany; horst.vollmar@ruhr-uni-bochum.de

**Keywords:** sepsis, post-intensive care syndrome (PICS), health care costs, aftercare, critical illness, primary care

## Abstract

**Background**: Survivors of sepsis often face long-term sequelae after intensive care treatment. Compared to the period of hospitalization, little is known about the ambulatory healthcare utilization in sepsis patients. The study evaluated healthcare utilization and associated costs of sepsis care including allied health professions after initial hospitalization. **Methods**: Secondary analysis was performed on data in 210 sepsis patients prospectively enrolled from nine intensive care study centers across Germany. Data was collected via structured surveys among their Primary care (Family-) physicians (PCPs) within the first month after discharge from ICU (baseline) and again at 6, 12 and 24 months after discharge, each relating to the period following the last survey. Costs were assessed by standardized cost unit rates from a health care system’s perspective. Changes in healthcare utilization and costs over time were calculated using the Wilcoxon rank-sum test. **Results**: Of the 210 patients enrolled, 146 (69.5%) patients completed the 24 months follow-up. In total, 109 patients were hospitalized within the first 6 months post-intensive care. Mean total direct costs per patient at 0–6 months were €17,531 (median: €6047), at 7–12 months €9029 (median: €3312), and at 13–24 months €18,703 (median: €12,828). The largest contributor to the total direct costs within the first 6 months was re-hospitalizations (€13,787 (median: €2965). After this first half year, we observed a significant decline in inpatient care costs for re-hospitalizations (*p* ≤ 0.001). PCPs were visited by more than 95% of patients over 24 months. **Conclusions**: Sepsis survivors have high health care utilization. Hospital readmissions are frequent and costly. Highest costs and hospitalizations were observed in more than half of patients within the first six months post-intensive care. Among all outpatient care providers, PCPs were consulted most frequently. **Clinical impact**: Sepsis survivors have a high healthcare utilization and related costs which persist after discharge from hospital. Within outpatient care, possible needs of sepsis survivors as physiotherapy or psychotherapy seem not to be met appropriately. Development of sepsis aftercare programs for early detection and treatment of complications should be prioritized.

## 1. Introduction

Sepsis is a life-threatening organ dysfunction due to infection [1]. As a global major health concern, it results in 15 to 19 million deaths per year [1]. Sepsis is known to be among the most costly causes of hospitalization [2,3,4,5], with estimates of annual costs totaling 62 billion dollars in the U.S. [6].

Surviving sepsis patients are confronted with long-term consequences of their critical illness, including new or worsened functional, cognitive, or psychiatric impairment and increased risk for medical deterioration [7]. Despite these long-term consequences of sepsis, there is no structured post-hospital rehabilitation program for sepsis survivors [7]. For this reason, multiple international initiatives and organizations such as the Executive Board of the World Health Organization [8] called for structured sepsis aftercare. Still, little is known about patients’ use of outpatient healthcare and related costs after hospital discharge, such as consultations, diagnostics, pharmacotherapy or supply with allied health services and durable medical equipment (DME) [9,10,11,12,13]. A literature review by Tiru et al. estimated that the initial inpatient costs account for only 30% of the sepsis-related costs, which also include productivity loss [1]. Multiple studies documented high rates of hospital readmissions and associated costs in intensive care unit (ICU) survivors [14,15,16,17]. Prescott et al. [18] showed that sepsis survivors experience increased inpatient utilization relative to their own pre-sepsis baseline. Hospital readmissions are frequent, costly, burdensome for patients and their families and a risk in terms of further infections. Mayr et al. [19] assessed that the total costs of 30-day readmissions for sepsis surpass those for chronic heart failure, acute myocardial infarction, pneumonia or chronic obstructive pulmonary disease. Most common causes of early readmission after severe sepsis are re-infections, with sepsis as leading readmission diagnosis [20].

To reach a better insight into the healthcare use and associated costs after initial hospitalization, the aim of this study in a German cohort of sepsis survivors was to evaluate the patterns and associated costs of healthcare utilization during the first 24 months after hospital discharge.

## 2. Materials and Methods

### 2.1. Study Design and Procedure

This is a secondary analysis of a prospectively followed sepsis survivor cohort [21]. Healthcare utilization and costs were collected during the first 24 months after discharge from ICU within the SMOOTH study (Sepsis survivors monitoring and coordination in outpatient health care). This randomized controlled clinical trial tested a primary care-intervention for sepsis survivors between 2011 and 2016 in Germany: 291 patients were recruited from nine ICUs across Germany. “Severe sepsis” or “septic shock” was identified according to the *ICD, tenth revision* codes (R65.1/R57.2) and American College of Chest Physicians/Society of Critical Care Medicine consensus criteria [22]. Data were derived from ICU documentation, primary care records and patient interviews. Patients´ clinical measures on depression, cognition, PTSD, neuropathy and pain were collected at baseline by validated instruments [23,24,25,26,27], as well as their health-related quality of life [28]—see Table 1, caption.

The intervention included training for both patients and their primary care physicians (PCPs) in evidence-based care and patient symptom monitoring including clinical decision support for the PCPs. Patients in the control group received usual care from their PCPs. Further details are published elsewhere [21,29,30]. During 24 months after ICU-discharge, there were no differences between intervention and control group regarding health care utilization, neither in inpatient nor in outpatient care (see [21], eTable 8, [30]). Thus, all 210 study participants with data available from primary care documentation were included in the present analysis.

### 2.2. Health Care Utilization

Patients’ PCPs were asked for all data on healthcare utilization that was accessible to them. The surveys occurred within the first month after discharge from ICU (baseline) and again at 6, 12 and 24 months after discharge, each relating to the period following the last survey. Specifically, PCPs were instructed to transfer all data on hospitalizations, rehabilitative care, specialist visits, clinical diagnostics as well as on prescription of allied health services, nursing care and medications from their documentation system to standardized questionnaires. In addition, PCPs were queried on patients’ nursing care and medications during the three months prior to index sepsis hospitalization. Furthermore, data on ICU treatment were collected, including information on the overall hospital length of stay (LOS), ventilation, renal replacement therapy as well as site and etiology of infection. Additionally, comorbidities were measured using the Charlson Comorbidity Index (CCI) [31].

### 2.3. Costs

Medication costs were calculated using standardized pharmacy selling prices from the “Lauer-Taxe” online-database 2016 from Lauer-Fischer GmbH, Fürth, Germany [32]. The “Lauer-Taxe” contains information on German-wide prices for drugs required to be sold through pharmacies. All other health care utilization costs were calculated using the standardized cost unit rates according to Bock et al. [33] and the EBM (“Einheitlicher Bewertungsmaßstab”) doctor’s reimbursement scheme provided by the National Association of Statutory Health Insurance Physicians (“Kassenaerztliche Bundesvereinigung”) Berlin, Germany [34]. The EBM contains German-wide rates for single and complex physician services (detailed description of the costing dataset in Appendix A). As there was no documentation regarding specification of nursing care, these costs were not calculated. Rehabilitation was valued and priced as inpatient treatment. In this study, costs were estimated in Euro based on payer perspective, so indirect costs, such as loss of productivity, absenteeism, or life-years lost, were not included. Costs were based on unit prices for the year 2011.

### 2.4. Main Measurements

We calculated health care utilization, total direct costs and costs separately by resource categories: hospital care, rehabilitation, PCP visits, outpatient specialist consultations, diagnostics (imaging and others), analgesic and antidepressant medication (as of special interest for sepsis aftercare), allied health visits (physical-, occupational-, speech- and podologist therapy) as well as Durable Medical Equipment (DME, for further description see Appendix A). The change in total direct healthcare costs between 0–6 and 13–24 months was determined to compare short- and long-term post-ICU resource utilization.

### 2.5. Analysis

We present descriptive statistics using mean ±standard deviation (SD), median and interquartile range (IQR), referring each to all patients with available data. We assessed differences in resource utilization and costs between 6–12 and 13–24 months using the Wilcoxon rank-sum test. All the statistical tests were two-sided, and *p*-values of <0.05 were considered as statistically significant. Patients, who died or dropped out for other reasons at the next follow-up, were excluded for the period concerned from analysis (per-protocol-analysis). Thus, all percentages are related to available data. Data preparation and analysis was conducted using IBM-SPSS-Statistics V20.0 (IBM Corp., Armonk, NY, USA).

## 3. Results

### 3.1. Patient Characteristics

Of 210 patients, the majority was male (68.1%) and mean age was 60.7 ± 14 years, see Table 1. Mean length of the initial ICU stay was 31.9 ± 27.2 days (median: 24 days, range 1–151 days); 84.7% (*n* = 177) of patients received mechanical ventilation, and 62.3% (*n* = 127) had a nosocomial infection as the cause of their sepsis.

At 6, 12, and 24 months, PCP data were available in 92.4% (*n* = 129), 84.3% (*n* = 177) and 69.5% (*n* = 146) of the study population, respectively, see Figure 1. In this period, 40 patients dropped out due to mortality, 25 data queries were rejected by the PCPs (mostly due to high workload), 10 patients did not visit their PCP within the inquiry period, and another 38 data queries failed for other reasons, mostly due to change of addresses.

### 3.2. Healthcare Use

In the six months after hospital discharge, 60% of the patients were readmitted at least once, spending a mean of 23.3 days hospitalized (median: 5.0 days). In contrast, from 7–12 and 13–24 months post-discharge, 49%/48% of patients were readmitted, spending a mean of 9.7/8.6 days in the hospital, (median: 0.00 days) (see Table 2).

Utilization of most healthcare categories declined from months 0–6 to months 13–24 post-ICU, even though data collection at 24 months was related to the prior 12 months, see Figure 2 and Appendix A.

In contrast, rates of medication use and nursing care, which were also available at three months before the sepsis event, increased remarkably until 6 months post ICU: We observed 12.5% of patients taking at least one new antidepressant and 7.8% of patients taking at least one new analgesic at 6 months compared to pre-sepsis, see Appendix A. The increased prescription rate persisted for up to 24 months after the initial hospital stay), in spite of a significant decrease of total medication costs between 0–6 and 13–24 month (*p* = 0.012), see Appendix A. At three months before sepsis, 16% of the patients required nursing care, compared to 56.7% at 6 months post-ICU, see Table 3.

### 3.3. Costs

Total direct costs per patient were (mean ± SD) €17,531 ± €23,954 (median: €6047) at 0–6 months, €9029 ± €12,708 (median: €3312) at 7–12 months and €18,703 ± €21,393 (median: €12,828) at 13–24 months, see Table 2. The largest contributors to total direct costs were re-hospitalizations (mean €13,787 ± €23,076; median: €2965) within the first 6 months, see Figure 2. During the first 6 months after ICU stay, 44% of the patients received rehabilitation treatment, with mean cost of €1591 ± €2196 (median: €0.00) per patient. Total costs for hospital and rehabilitation care declined from months 0–6 to months 13–24, *p* = 0.001, see Appendix A.

Specialist visits accounted for more than half of total outpatient visit costs up to 6 months after discharge, see Table 2 and Appendix A. Medications, diagnostic imaging, allied health visits, and DME represented minor costs accounting for less than 3% of direct total costs.

## 4. Discussion

This study reports on the comprehensive health care utilization and associated costs of 210 sepsis survivors up to 24 months after discharge from ICU. Results showed that about half of the patients were hospitalized within the first 6 months post-intensive care. Mean total direct costs per patient in this period were €17,531 compared to €9029 in the second half year after discharge (in total €26,559 in the first year), and €18,703 during the second year after discharge. The largest contributor to the total direct costs in the first 6 months was re-hospitalizations (€13,787 of the €17,531 total direct costs). After this first half year, there was a significant decline in inpatient care costs for re-hospitalizations. PCPs were visited by more than 95% of patients over 24 months.

Our cohort is comparable to other sepsis survivor cohorts in terms of age and comorbidities [35,36], but had a higher rate of exposure to mechanical ventilation and longer duration of ICU length of stay than other studies [9,12,15,16,17,18,21,37]. Our cohort experienced the highest health care utilization and costs during the first 6 months after ICU discharge, which roughly halved in the six months thereafter. This is consistent with prior studies [9,12,15,16,17,18,37]. Both inpatient and outpatient utilization were considerably higher compared to the general German population [38]. This may reflect sepsis-related needs as well as already higher health care utilization in our cohort before sepsis – resulting from pre-existing comorbidities (see Table 1). Considering indirect economic impact such as productivity loss or need for informal nursing care, total financial toll of sepsis needs to be estimated even higher. High standard deviations of obtained results reflect high heterogeneity of patient’s resource consumption and care conditions.

Taking into account differences between cohorts and international health care systems, it can be observed that the mean total direct costs of €26,559 per patient for the first year after ICU discharge of our cohort were comparable to an extended U.S. health-insurance claims database including 16,019 patients with severe sepsis, totaling $33,900 [39]. A recent retrospective Canadian cohort study using administrative data found even higher total mean costs (CAN$ 66,781) in 64,204 severe sepsis survivors [40] with a mean age of 74 years and a mean ADG Score of 34. In contrast, other studies resulted in lower mean total costs for the same period, as

-another U.S. claims data analyses of *n* = 2834 patients with severe sepsis (totaling $18,425) [41]-two prospectively followed cohorts of *n* = 502 Canadian severe sepsis survivors (totaling CAN$ 20,855) [42] and *n* = 839 U.S. ARDS (acute respiratory distress syndrome) survivors (totaling $23,651, including $6761 outpatient costs) [43] and-a recent extensive German claims data analysis among 116,507 sepsis survivors (totaling €14,891 [44].

Baseline characteristics such as age [41], percentage of ICU-treatment [41], mean Charlson Comorbidity Index [44] and APACHE II-Score [42] suggest lower disease severity of these cohorts.

### 4.1. Inpatient Care

Consistent with prior studies [15,16,18,37,39,41,42,45] there was substantial inpatient utilization in the first 6 months, which accounted for the major part of total costs. The readmission rate in our cohort within the first 6 months was with 60% above the mean rate of 36.2 % extracted from seven studies with a total of *n* = 107,293 sepsis patients, according to a recent meta-analysis [46]. In contrast, 90% of a prospectively followed German cohort of *n* = 396 ARDS-survivors were rehospitalized in the first year after discharge from ICU [47].

According to literature, most common reasons for rehospitalizations were (re-) infections or exacerbation of existing diseases [7,45,46]. A current retrospective observation study following sepsis found pneumonia, exacerbation of congestive heart failure, chronic obstructive pulmonary disease, acute renal failure, and urinary tract infection as the most common diagnoses for hospital readmissions [20]. The estimation, that more than one third of readmissions may be preventable [20] highlights the need for a structured sepsis aftercare.

Even though only 44% of patients were admitted to rehabilitation treatment after ICU discharge, it represented the second highest cost factor in our cohort. In the United States only 32% of ARDS survivors had a rehabilitation or skilled nursing facility stay within the first year after discharge [9]. In Taiwan only 14% of sepsis survivors received rehabilitation post-ICU [48]. Differences may be attributed to different healthcare systems as well. However, the overall impact of rehabilitation on disease outcome after critical illness is not clearly established: a systematic review from Taito et al. found no evidence on decreased mortality and low evidence on increased quality of life compared to standard care [49]. This could possibly show that rehabilitation facilities are currently not sufficiently prepared for the specific needs of critical illness survivors indicating the lack of structured sepsis specific rehabilitation programs and guidelines [50,51].

### 4.2. Outpatient Care

Within our cohort, outpatient costs made up only less than a fifth of total health care utilization following ICU treatment, compared to half of the total direct costs reported from U.S. claims data analysis [41]. However, in the second year after the initial hospitalization, inpatient utilization decreases, whereas proportion of outpatient utilization increases.

Nearly all documented patients visited both PCPs and specialists. PCPs were consulted most frequently-every third week up to 6 months and every fourth week up to 24 months after ICU discharge. While >95% of sepsis survivors in our cohort saw a PCP on average more than once a months, only 83% of the general German population in the same age group visited a PCP at least once a year [38]. (This finding however may be in part attributable to the PCP-centered intervention in the SMOOTH study.) A population-based study from northeastern Germany [12] showed ICU treatment being as well associated with a higher number and costs of annual outpatient visits: Specifically, specialist visits (internal medicine, surgery, orthopedics and psychiatry) were increased, whereas there was no significant effect on PCP visits.

In the general German population gynecologists (for women) and ophthalmologists are most frequently consulted [38]. In contrast, we observed 53% of sepsis survivors with at least one contact to internists compared to 35% within the general German population. These numbers may reflect the high proportion of intern ailment causing possible sequelae in sepsis survivors, see measures of comorbidity at Table 1. Similar consultation rates of PCPs and internists is shown by self-report of *n* = 396 ARDS-survivors in Germany [13]. Frequent contacts to urologists may be associated with a higher proportion of males and an older age in the study cohort, as urological check-ups are usual in this subgroup [52]. The utilization of outpatient surgical care could be justified by the high rate of sepsis caused around surgical procedures (see Table 1). With regard to the prevalence of posttraumatic stress disorder (14.6%) and depression (24.2%) after discharge from ICU within the cohort (see Table 1), utilization of neurologic/psychiatric specialist care seems to be low. In contrast, a national US multicenter study investigating survivors of ARDS found psychiatrists to be the most frequently visited specialists [9]. Our results may be attributable to the long waiting time for psychotherapy in Germany [53]. Accordingly, the increase of treatment with at least one antidepressant at 6 months after ICU discharge (see Appendix A) might indicate a stronger focus on pharmacotherapy rather than psychotherapy, mainly managed by PCPs [53,54]. In a Danish study on nonsurgical critically ill patients receiving mechanical ventilation, newly established psychiatric diagnoses and psychopharmacological treatment were increased in the first three months after discharge [10].

The large number of involved specialists and hospital readmissions might be a complicating factor regarding efficient coordination of sepsis aftercare.

### 4.3. Nursing

The substantial increase in need for nursing between 3 months before and 6 months after sepsis (see Table 3) indicates the burden of disease of both sepsis survivors and their relatives as well the socioeconomic impact on society. Our findings go along with the recent German claims data analysis by Fleischmann-Struzek et al., finding 31.5% (95%CI, 31.1–31.8%) new nursing care dependency during the first year post sepsis [44,54]. Within the U.S. Nationwide Inpatient Sample (NIS), Kumar et al. demonstrated an increasing need of nursing care after sepsis due to higher survival rates with one third of survivors of severe sepsis being discharged to skilled nursing facilities [55]. A secondary analyses of international severe sepsis survivors who lived independently at home prior sepsis resulted in a similar increase with 41.6% of patients who could not live independently 6 months after sepsis [56].

### 4.4. Allied Health Services

So far, there are barely investigations regarding the usage of allied health services after sepsis. As with rehospitalisations, rehabilitation, and medication costs: The utilization of allied health services—mostly physiotherapy followed by occupational therapy—was higher at 6 months than at 24 months (see Table 2). Taking the high degree of physical impairment at discharge into account (SF-36 PCS mean 25.3 ± 8.8) and 59.2% prevalence of neuropathy [21,29], overall usage seems low with 60% of patients receiving any such treatments at 6 months after sepsis.

As shown by Zhang et al. and others, early rehabilitation and physiotherapy increases muscle strength, physical function, and quality of life [57]. Thus, low utilization of physiotherapy was identified as a major lack in sepsis aftercare, which should be addressed in future PCP training.

In all categories of health care utilization and related costs high statistical variation was observed. Both heterogeneity in patients’ comorbidity and lack of medical guidelines for sepsis aftercare may contribute to heterogeneous diagnostic and therapeutic decisions PCPs have to take, which might contribute to highly varied patterns of health care usage.

### 4.5. Strengths and Limitations

This study examined the utilization and related costs of different categories of health care in a cohort of sepsis survivors in Germany over an observation period of two years after ICU discharge. To our knowledge, a differentiated evaluation of post-sepsis outpatient aftercare in various health care services including PCPs and specialists, medication, imaging diagnostics, allied health services, DME and nursing, based on prospectively collected primary data has not been done before. Furthermore, data obtained from the patient’s PCPs add more details and accuracy to patient-reported outcomes. As a result, higher significance may be reached when cross checking against data from other sources, such as claims data.

Our study has several limitations: First, no control group with a similar level of morbidity but without sepsis was available to compare resource utilization and to measure costs specifically attributable to sepsis. Pre-sepsis comparison data regarding resource utilization could only be obtained for medication and nursing. Our study does not give information on the diagnoses for hospital readmissions and indications for outpatient service utilization. Even if there was no statistical difference found between the intervention group and control group in the parent study, data were collected within a RCT on sepsis aftercare, which may have influenced resource use. Consequently, the impact of other factors than sepsis-survival on resource utilization cannot be estimated.

Second, not all possible health care costs were available to PCP documentation, primary care records are characterized by a high number of missing values. Due to missing specification, costs of nursing care were not included. Standardized cost unit rates for inpatient services were applied for cost calculations, which are only a rough estimate. Thus, significance of summed up total direct costs is limited.

Third, patients who had died or left the study for other reasons before the next follow-up were not included in this analysis, which may have produced a selection bias. In light of differences in baseline characteristics to other cohorts [58] and limited comparability of Sepsis-2 and the current Sepsis-3 definition [59], our results may not be generalizable for all sepsis survivors. In addition, international differences between health care systems are still high. Summarizing, comparability with other studies is limited.

## 5. Conclusions

Our study found high health care utilization and associated costs in a German cohort of sepsis survivors. Total costs were highest at 0–6 months post-intensive care. After the first 6 months costs decreased by almost half and stayed stable. More than patients and families, long-term burden of sepsis does also affect the healthcare system. We were able to confirm hospital readmission as being the major cost factor, followed by rehabilitation treatment. Within outpatient care, several possible specific needs of sepsis survivors, such as neuropsychological/psychiatric specialist care or physiotherapy seem to be undersupplied. Thus, further development of sepsis aftercare programs for early detection and treatment of complications may be beneficial.

## Figures and Tables

**Figure 1 jcm-11-01142-f001:**
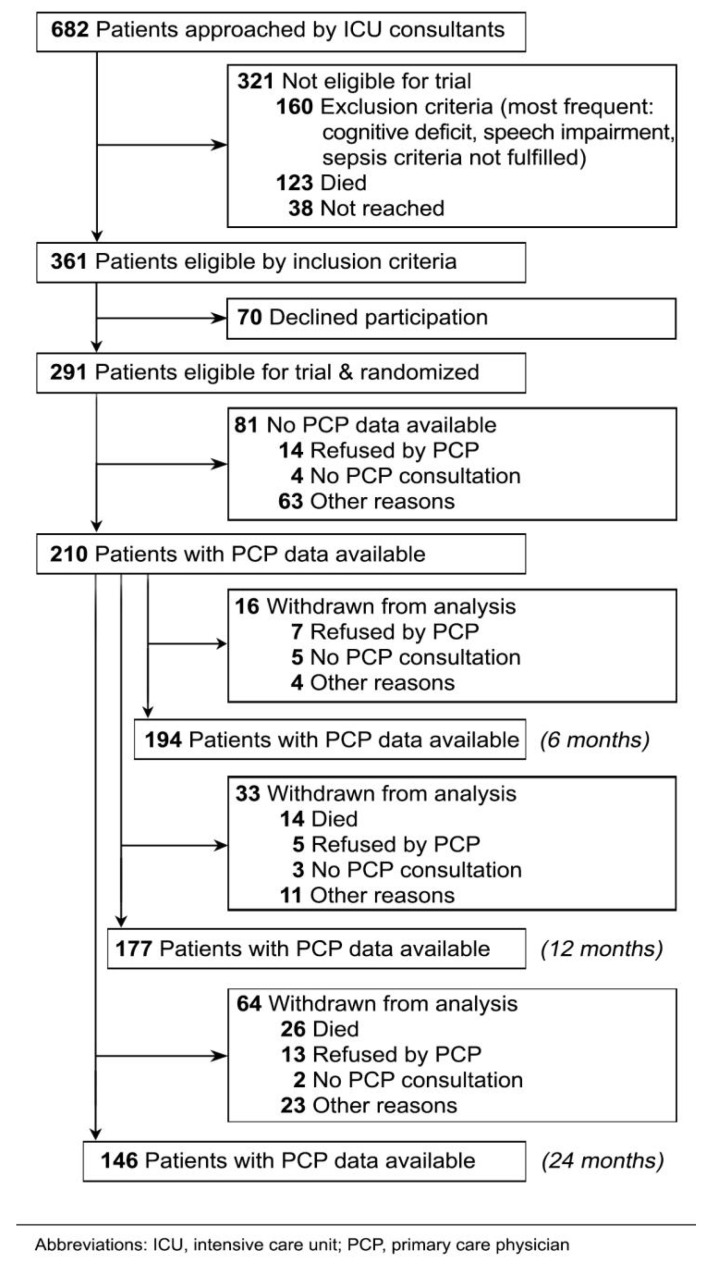
CONSORT Flow Chart of patient recruitment and retention.

**Figure 2 jcm-11-01142-f002:**
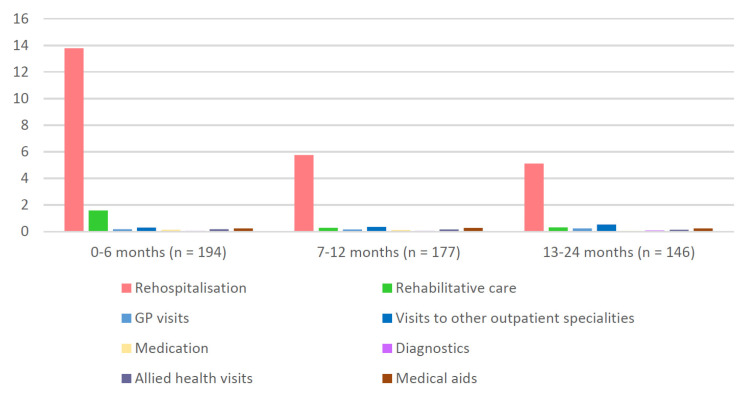
Cost categories by follow-up period. Mean values in thousand Euro (k€) are given.

**Table 1 jcm-11-01142-t001:** Baseline characteristics of the study population.

		All (*n* = 210)	NA
Sociodemographics			
	Age, years, mean (SD)	60.7 (14.0)	0
	Male, (*n* (%))	143 (68.1)	0
	Married, (*n* (%))	109 (51.9)	2
	Educational status “< High school”, (*n* (%))	72 (34.3)	0
Care measures			
	Recent surgical history, (*n* (%))		2
	Emergency	76 (36.5)	
	Elective surgery	47 (22.6)	
	No history	47 (22.6)	
	ICU length of stay days: median, mean (SD) (Q1;Q3)	24, 31.9 (27.2) (13; 43.2)	16
	Mechanical ventilation, (*n* (%))	177 (84.7)	1
	if applicable, days: median, mean (SD) (Q1;Q3)	8, 14.6 (18.92) (2; 21.5)	9
	Renal replacement therapy, (*n* (%))	61 (29.3)	2
	if applicable, days: median, mean (SD) (Q1;Q3)	0, 3.26 (8.5) (0; 2)	7
Clinical Measures			
	Source of infection, (*n* (%))		6
	Community acquired	77 (36.7)	
	Nosocomial (ICU or IMC)	106 (52.0)	
	Nosocomial (general ward or nursing home)	21 (10.3)	
	Site of infection (*n* (%)) ^#^		3
	Pneumonia	89 (43)	
	intraabdominal	32 (15.5)	
	respiratory tract, other	25 (12)	
	urogenital	20 (9.7)	
	bone, soft tissue	19 (9.2)	
	cardiovascular	13 (6.3)	
	Comorbidity: Charlson Index ^a,^*, mean (SD)	3.83 (3.0)	1
	ICD-diagnoses, No., median, mean (SD)	9, 9.5 (5.1)	0
	BMI, mean (SD)	27.5 (5.9)	6
	Depression, MDI ^b,^*, mean (SD)	17.7 (9.9)	1
	PTSD, PTSS-10 ^c,^*, mean(SD)	23.1 (9.8)	1
	Cognition: TICS-M ^b,f,†^, mean (SD)	33.7 (3.4)	0
	Neuropathic symptoms: NSS ^d,^* mean (SD)	3.6 (3.2)	4
	Score 3–10, (*n* (%))	123 (58.6)	
	Pain		
	Intensity: GCPS PI ^e,^* mean (SD)	43.3 (24.8)	3
	Disability: GCPS DS ^e,^* mean (SD)	36.3 (34.6)	5
Quality-of-Life measures			
	SF-36 MCS ^e,†^, mean (SD)	49.0 (12.7)	4
	SF-36 PCS ^e,†^, mean (SD)	26.0 (8.9)	15

Abbreviations: BMI, Body Mass Index; GCPS DS, Graded Chronic Pain Scale Disability Score [27]; GCPS PI, Graded Chronic Pain Scale Pain Intensity [27]; ICU, intensive care unit; IMC, Intermediate Care; MDI, Major Depression Inventory [23]; NA, Not Available; NSS, Neuropathic Symptom Score [26]; PTSD, Post Traumatic Stress Disorder; PTSS, Post-Traumatic Symptom Scale [28]; SF-36 MCS, Short Form (36) Health Survey Mental Component Score [28]; SF-36 PCS, Short Form (36) Health Survey Physical Component Score [28]; TICS-M, modified Telephone Interview for Cognitive Status [24]; i, intervention; c, control. Anchors: * high score indicates high impairment, ^†^ high score indicates low impairment. Ranges: ^a^ The range of possible scores is 0–37, ^b^ The range of possible scores is 0–50, ^c^ The range of possible scores is 10–70, ^d^ The range of possible scores is 0–10, ^e^ The range of possible scores is 0–100, ^f^ values only above 27 (inclusion criteria). ^#^ main six sites of infection are shown.

**Table 2 jcm-11-01142-t002:** Subcategories of resource use and healthcare costs post-sepsis over time.

	0–6 months (*n* = 194)					7–12 months (*n* = 177)					13–24 months (*n* = 146)				
	*n* *	Mean	SD	Median	Percentiles (25;75)	*n* *	Mean	SD	Median	Percentiles (25;75)	*n* *	Mean	SD	Median	Percentiles (25;75)
**Rehospitalisation:** **Costs per patient**^a^ (€)	**182** **(109)**	**13,786.6**	**23,075.86**	**2965.20**	**0.00;** **17,791.20**	**166** **(81)**	**5755.35**	**10,352.6**	**0.00**	**0.00;** **6968.22**	**133** **(64)**	**5114.41**	**10,580.9**	**0.00**	**0.00;** **5930.40**
Rehospitalisation: total number of hospital readmissions	182 (109)	1.29	1.61	1.00	0.00; 2.00	166 (81)	0.96	1.33	0.00	0.00; 1.25	133 (64)	0.71	1.15	0.00	0.00; 1.00
Rehospitalisation: length of stay (d)	184 (109)	23.25	38.91	5.00	0.00; 30.00	167 (82)	9.70	17.46	0.00	0.00; 11.75	135 (66)	8.62	17.84	0.00	0.00; 10.00
**Rehabilitative Care:** **Costs per patient**^a^ (€)	**183** **(81)**	**1591.37**	**2195.46**	**0.00**	**0.00;** **2680.70**	**116** **(11)**	**283.62**	**913.65**	**0.00**	**0.00;** **0.00**	**136** **(13)**	**311.79**	**1053.51**	**0.00**	**0.00;** **0.00**
Rehabilitative Care: length of stay (d)	183 (81)	13.06	18.02	0.00	0.00; 22.00	116 (11)	2.33	7.50	0.00	0.00; 0.00	137 (20)	2.56	8.65	0.00	0.00; 0.00
**Costs per patient** **(GP visits)**^a^ (€)	**184** **(179)**	**184.37**	**159.93**	**140.42**	**80.24;** **240.72**	**165** **(160)**	**169.60**	**115.48**	**140.42**	**100.30;** **220.66**	**137** **(133)**	**238.67**	**210.03**	**180.54**	**110.33;** **300.90**
Number of GP visits per patient	184 (179)	9.69	7.89	8.00	5.00; 12.00	165 (160)	8.45	5.76	7.00	5.00; 11.00	137 (133)	11.90	10.47	9.00	5.50; 15.00
**Costs per patient** **(all other specialities)**^a^ (€)	**185**	**305.90**	**467.16**	**130.88**	**24.70;** **345.52**	**167**	**354.60**	**553.44**	**171.25**	**65.44;** **363.32**	**138**	**537.32**	**1057.99**	**173.92**	**25.42;** **578.22**
Number of physician visits per patient (all other specialities)	177	6.29	9.57	2.00	1.00; 6.50	156	7.14	10.59	4.00	1.00; 8.00	130	10.75	22.53	4.00	1.00; 12.00
Costs per patient (Internal Medicine) ^a^ (€)	(99)	195.61	341.88	65.44	0.00; 261.76	(103)	222.57	400.71	65.44	0.00; 196.32	(80)	348.54	661.86	65.44	0.00; 327.20
Costs per patient (Neurology/Psychiatry) ^a^ (€)	(30)	24.17	66.75	0.00	0.00; 0.00	(30)	29.19	75.87	0.00	0.00; 0.00	(18)	39.54	155.77	0.00	0.00; 0.00
Costs per patient (Surgery) ^a^ (€)	(28)	25.33	69.27	0.00	0.00; 0.00	(27)	31.70	98.24	0.00	0.00; 0.00	(20)	27.35	88.36	0.00	0.00; 0.00
Costs per patient (Dermatology) ^a^ (€)	(9)	4.90	23.53	0.00	0.00; 0.00	(14)	5.32	19.88	0.00	0.00; 0.00	(11)	7.80	32.97	0.00	0.00; 0.00
Costs per patient (Urology) ^a^ (€)	(25)	11.48	35.03	0.00	0.00; 0.00	(20)	13.16	43.87	0.00	0.00; 0.00	(16)	17.72	73.22	0.00	0.00; 0.00
Costs per patient (Gynaekology) ^a^ (€)	(7)	6.03	33.36	0.00	0.00; 0.00	(6)	4.69	33.40	0.00	0.00; 0.00	(4)	3.06	22.53	0.00	0.00; 0.00
Costs per patient (ENT) ^a^ (€)	(16)	10.13	38.03	0.00	0.00; 0.00	(14)	9.33	35.02	0.00	0.00; 0.00	(8)	12.65	72.09	0.00	0.00; 0.00
Costs per patient (Ophthalmology) ^a^ (€)	(22)	18.99	61.42	0.00	0.00; 0.00	(22)	21.03	64.22	0.00	0.00; 0.00	(15)	28.48	118.58	0.00	0.00; 0.00
Costs per patient (Orthopaedics) ^a^ (€)	(17)	8.66	31.67	0.00	0.00; 0.00	(17)	8.52	32.13	0.00	0.00; 0.00	(23)	21.55	76.96	0.00	0.00; 0.00
Costs per patient (Psychotherapists) ^a^ (€)	(1)	3.80	51.67	0.00	0.00; 0.00	(4)	7.95	58.53	0.00	0.00; 0.00	(4)	15.84	103.41	0.00	0.00; 0.00
**Medication costs per patient**^b^ (€)	**194**	**143.66**	**383.19**	**0.00**	**0.00;** **65.25**	**175**	**119.83**	**304.86**	**0.00**	**0.00;** **99.00**	**146**	**48.65**	**235.35**	**0.00**	**0.00;** **0.00**
Antidepressants: number of prescribed drugs per patient	194 (37)	0.20	0.43	0.00	0.00; 0.00	177 (30)	0.21	0.51	0.00	0.00; 0.00	146 (4)	0.03	0.22	0.00	0.00; 0.00
Antidepressants: Costs per patient ^b^ (€)		17.76	56.07	0.00	0.00; 0.00		19.71	59.16	0.00	0.00; 0.00		2.12	14.15	0.00	0.00; 0.00
Analgesics: number of prescribed drugs per patient	194 (33)	0.43	0.81	0.00	0.00; 1.00	175 (44)	0.36	0.69	0.00	0.00; 1.00	164 (10)	0.11	0.46	0.00	0.00; 0.00
Analgesics: Costs per patient ^b^ (€)		125.91	376.94	0.00	0.00; 45.00		99.90	292.37	0.00	0.00; 45.00		46.53	234.44	0.00	0.00; 0.00
**Diagnostics: Costs per Patient** **(all diagnostics)**^c^ (€)	**194**	**59.62**	**125.52**	**0.00**	**0.00;** **65.95**	**177**	**59.43**	**134.01**	**0.00**	**0.00;** **59.64**	**146**	**84.39**	**161.27**	**9.08**	**0.00;** **116.47**
Diagnostics: Number of diagnostical methods (imaging + non-imaging)	194	1.29	2.49	0.00	0.00; 2.00	177	1.07	1.85	0.00	0.00; 2.00	146	1.77	2.82	1.00	0.00; 2.00
Imaging methods: number of used diagnostical methods	194 (73)	0.95	1.98	0.00	0.00; 1.00	177 (66)	0.81	1.49	0.00	0.00; 1.00	146 (73)	1.47	2.39	0.50	0.00; 2.00
Imaging methods: Costs per patients ^c^ (€)		44.13	108.00	0.00	0.00; 31.72		39.26	115.38	0.00	0.00; 28.07		62.93	119.67	3.86	0.00; 72.13
Other diagnostics: number of used diagnostical methods	194 (33)	0.34	1.24	0.00	0.00; 0.00	177 (33)	0.26	0.67	0.00	0.00; 0.00	146 (23)	0.30	0.96	0.00	0.00; 0.00
Other diagnostics: Costs per patient ^c^ (€)		15.49	49.21	0.00	0.00; 0.00		20.17	64.83	0.00	0.00; 0.00		21.46	72.79	0.00	0.00; 0.00
**Allied health visits** **Costs per patient** **(all visits)**^a^ (€)	**143** **(86)**	**177.90**	**218.26**	**164.20**	**0.00;** **164.20**	**136** **(77)**	**171.30**	**226.96**	**164.20**	**0.00;** **164.20**	**112** **(53)**	**139.88**	**209.30**	**0.00**	**0.00;** **164.20**
Number of visits per patient	143 (86)	0.81	0.80	1.00	0.00; 1.00	136 (77)	0.78	0.85	1.00	0.00; 1.00	112 (53)	0.63	0.77	0.00	0.00; 1.00
Physiotherapy: Costs per patient (€)	133 (81)	103.71	86.87	164.20	0.00; 164.20	132 (73)	97.03	92.87	164.20	0.00; 164.20	118 (50)	73.89	82.10	0.00	0.00; 164.20
Occupational Therapy: Costs per patient (€)	139 (25)	67.69	144.79	0.00	0.00; 0.00	138 (19)	53.99	132.17	0.00	0.00; 0.00	104 (14)	47.74	125.58	0.00	0.00; 0.00
**Medical aids:** **Costs per patient**^a^ (€)	**143** **(66)**	**234.12**	**503.67**	**0.00**	**0.00;** **193.40**	**136** **(59)**	**280.83**	**612.83**	**0.00**	**0.00;** **346.43**	**112** **(46)**	**230.36**	**538.39**	**0.00**	**0.00;** **140.66**
Number of medical aids per patient	143 (66)	0.73	1.01	0.00	0.00; 1.00	136 (59)	0.79	1.21	0.00	0.00; 1.00	112 (46)	0.77	1.16	0.00	0.00; 1.00
**Total direct costs** **per patient**^a,b,c,#^ (€)	**131**	**17,530.6**	**23,953.8**	**6047.40**	**1718.21;** **22,398.59**	**80**	**9028.77**	**12,708.1**	**3311.69**	**1021.35;** **11,583.41**	**15**	**18,703.1**	**21,393.1**	**12,828.4**	**7152.41;** **26,426.30**

* no. of patients with available data (no. of patients with at least one corresponding healthcare utilization). ^#^ due to different number of available patients, subcategories do not sum up to total costs. ^a^ All costs assessed by standardized cost unit rates according to Bock et al. [33]; ^b^ All medication costs from LAUER-Taxe online-database [32]; ^c^ Costs for diagnostics the EBM (“Einheitlicher Bewertungsmaßstab”)-reimbursement scheme provided by the “Kassenaerztliche Bundesvereinigung“ (National Association of Statutory Health Insurance Physicians), Berlin, Germany [34] Main cost categories are printed in bold. Regarding outpatient treatment within the first 6 months post-discharge, 97% of patients visited a PCP. PCPs were consulted more frequently during months 0–6 than during months 7–24 after ICU discharge. Internal medicine specialists were consulted by 54% of patients, with a mean of six visits.

**Table 3 jcm-11-01142-t003:** Comparison of nursing care at three months before sepsis and six months post-ICU. (*n* = 194 Patients).

	3 Months before Sepsis	6 Months Post-ICU
Number of patients receiving nursing care, No. (%)	31 (16.0%)	110 (56.7%)
Proportion of patients receiving inpatient nursing care, No. (%)	2 (1.0%)	8 (4.1%)
Proportion of patients receiving home care (with day-stationary care), No. (%)	18 (9.3%)	39 (20.1%)
Proportion of patients receiving informal care, No. (%)	18 (9.3%)	37 (19.1%)
Patients with any approved nursing care need *, No. (%)	20 (10.4%)	58 (29.9%)
Nursing care level I ^a^, No. (%)	10 (5.2%)	35 (18.0%)
Nursing care level II ^b^, No. (%)	6 (3.1%)	15 (7.7%)
Nursing care level III ^c^, No. (%)	4 (2.1%)	5 (2.6%)

Numbers don’t add up to 100% due to overlaps of dimensions. * Need for approved nursing care has been assessed by the Medical Review Board of the Statutory Health Insurance Funds (MDK). ^a^ Need for support in body care, mobility or nutrition once a day. ^b^ Nursing care level II: Need for support in body care, mobility or nutrition three times a day. ^c^ Nursing care level III: Need for nursing day and night.

## Data Availability

The datasets used and/or analysed during the current study are available from the corresponding author on reasonable request.

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
