# Peer review of "Healthcare Utilization and Costs in Sepsis Survivors in Germany–Secondary Analysis of a Prospective Cohort Study"

_jcm, 2022, doi:10.3390/jcm11041142_

Round 1
Reviewer 1 Report
This article aims to report the health care cost after ICU discharge for the sepsis or septic shock. It is a post hoc analysis of a german RCT. The cost for each patient is assessed on a 24 month period and very well documented. My main concern is the reporting of the results:
- Concerning the estimates: mean and median are reported, but very different in favor of their non linear distrubution, rendering interpretation of the results confusing: only median should be reported.
- comparisons between periods are not reported in the table;
- Most of the results are reported in table 2. I believe that a figure in the manuscrit is needed to see immediatly the distribution of the source of the costs.
- comparison of the cost between survivors and no survivors would also have been interesting.
Reviewer 2 Report
In the paper “Healthcare utilization and costs in sepsis survivors in Germany– a secondary analysis of a prospective cohort study” the authors evaluated healthcare utilization and associated costs of sepsis care in Germany, including allied health professions’ after initial hospitalization.
The study represents secondary analysis that was performed on data in 210 sepsis patients prospectively enrolled from nine intensive care study centers across Germany. The data are very informative with some novel aspects and analysis of health care utilization in sepsis survivors. They have shown that hospital re-admissions are frequent and costly. Highest costs and hospitalizations were observed in more than half of patients within the first six months post intensive care. Among all outpatient care providers, primary care physicians were consulted most frequently. Study does have limitations but they are listed in the manuscript. Very valuable analysis and will be very interesting to expand the study and compare results with similar analysis in other countries.
Comments:
- The description of some methods used is missing. They should describe the methods used in Table 1 for baseline characteristics of the study population in the Supplementary material or provide the links for the forms used, i.e. for TICS-M, modified Telephone Interview for Cognitive Status; pain intensity; cognition, etc.
- Figure E1A and B should be part of the main manuscript, not supplements. It’s very informative and summarizes the findings of the analysis.
- Row 157, in the brackets it’s written: Error! Reference source not found!
